# A Chronological Study on Grapevine Leafroll-Associated Virus 2 in Australia

**DOI:** 10.3390/v15051105

**Published:** 2023-04-30

**Authors:** Nuredin Habili, Qi Wu, Amy Rinaldo, Fiona Constable

**Affiliations:** 1The Australian Wine Research Institute, Wine Innovation Central Building, Hartley Grove, crn Paratoo Road, Urrbrae, SA 5064, Australia; 2School of Agriculture, Food and Wine, University of Adelaide, Waite Precinct, PMB 1, Glen Osmond, SA 5064, Australia; 3GRDC, 187 Fullarton Road, Dulwich, SA 5065, Australia; 4AgriBioscience, Agriculture Victoria’ Department of Energy, Environment and Climate Action, AgriBio, 5 Ring Road, Bundoora, VIC 3083, Australia; 5School of Applied Systems Biology, La Trobe University, Bundoora, VIC 3083, Australia

**Keywords:** grapevine leafroll-associated virus 2, *Closterovirus*, graft incompatibility, hypersensitive reaction, RNA silencing suppressor

## Abstract

Grapevine leafroll disease affects the health status of grapevines worldwide. Most studies in Australia have focused on grapevine leafroll-associated viruses 1 and 3, while little attention has been given to other leafroll virus types, in particular, grapevine leafroll-associated virus 2 (GLRaV-2). A chronological record of the temporal occurrence of GLRaV-2 in Australia since 2001 is reported. From a total of 11,257 samples, 313 tested positive, with an overall incidence of 2.7%. This virus has been detected in 18 grapevine varieties and *Vitis* rootstocks in different regions of Australia. Most varieties were symptomless on their own roots, while Chardonnay showed a decline in virus-sensitive rootstocks. An isolate of GLRaV-2, on own-rooted *Vitis vinifera* cv. Grenache, clone SA137, was associated with severe leafroll symptoms after veraison with abnormal leaf necrosis. The metagenomic sequencing results of the virus in two plants of this variety confirmed the presence of GLRaV-2, as well as two inert viruses, grapevine rupestris stem pitting-associated virus (GRSPaV) and grapevine rupestris vein feathering virus (GRVFV). No other leafroll-associated viruses were detected. Among the viroids, hop stunt viroid and grapevine yellow speckle viroid 1 were detected. Of the six phylogenetic groups identified in GLRaV-2, we report the presence of four groups in Australia. Three of these groups were detected in two plants of cv. Grenache, without finding any recombination event. The hypersensitive reaction of certain American hybrid rootstocks to GLRaV-2 is discussed. Due to the association of GLRaV-2 with graft incompatibility and vine decline, the risk from this virus in regions where hybrid *Vitis* rootstocks are used cannot be overlooked.

## 1. Introduction

Leafroll is a major grapevine disease with a worldwide distribution. It is associated with major losses to all sectors of the vine and wine industries [1], and is considered to be the most important virus disease of the grapevine in Australia [2]. Six species of the *Closteroviridae* family are associated with grapevine leafroll disease. Four of these, i.e., grapevine leafroll-associated virus 1 (GLRaV-1), GLRaV-3, -4, and -13 are in the genus *Ampelovirus*. GLRaV-13 is the latest addition to the genus, which was identified in *Vitis vinifera* cv. Koshu, in Japan, but its symptom expression status is not known since it was co-infected with other leafroll viruses [3]. GLRaV-2 and GLRaV-7 are assigned in the genera *Closterovirus* and *Velarivirus*, respectively [4,5]. GLRaV-1, -2, -3 and -4 occur in Australian vineyards but GLRaV-7 and -13 have not been reported [2].

GLRaV-2 is the only *Closterovirus* species in the *Closteroviridae* family that infects grapevine. The virus was characterised as grapevine leafroll-associated virus 2 in 1990 [6]. The naming was later confirmed by comparative biological and serological studies in Italy [7] and France [8], showing unique properties compared to other leafroll viruses. GLRaV-2 is the only leafroll virus that is mechanically transmissible to certain herbaceous hosts, but not grapevine. It is one of the *Closterovirus* species for which no vector is known.

In contrast to ampeloviruses, which are believed to have originated from the Old World [9], it has been hypothesised that GLRaV-2 emerged in North America, as it has been detected in native American *Vitis* species [10]. However, the virus has also been detected in own-rooted autochthonous *V. vinifera* varieties in Iran, where the vines have been grown on their own roots for thousands of years without being grafted to native American *Vitis* species [11].

Two types of symptoms have been reported in GLRaV-2-infected vines. Firstly, leafroll symptoms, which are generally considered to be mild and often absent in white varieties [12]. These are characterised by the downward rolling of leaf margins and reddening of leaves in red varieties, while no symptoms are visible in own-rooted white varieties [12]. The second type of symptom is graft incompatibility (GI), which is associated with disorders developed in sensitive rootstocks following their grafting. Unfortunately, the range of symptoms associated with GLRaV-2 has not been well studied, mainly due to the problem of mixed infections with other leafroll viruses.

GLRaV-2 variants have diverse genomes, which can show nucleotide sequence variability between 25 and 30%, based on the 70-kDa heat-shock protein homolog (HSP70h) and major capsid protein (CP) genes [13]. One phylogroup (RG) was initially considered to be a distinct species (grapevine rootstock stem lesion-associated virus), but it is now included in GLRaV-2 [12,14]. 

The variants of GLRaV-2 are scattered among six phylogenetic groups based on the analysis of the CP and HSP70h nucleotide sequences [9,13], which to some extent, reflect their pathogenic properties. These are PN, 93/955, H4, BD, RG and PT20. Members of the PN and 93/955 groups are associated with leafroll symptoms and GI depending on the viral isolate. The BD group appears to be asymptomatic and is rarely associated with leafroll symptoms or GI [12,13]. The RG group is associated with GI symptoms which only develop following grafting, but not when the vines are on their own roots [15]. 

In Australia, GLRaV-2 was first detected in cv. Black July, a table grape, with leafroll symptoms [16]. An initial report on the incidence of GLRaV-2 in commercial samples up to the year 2000 is available (61 samples tested positive from a total of 2479 samples, with an incidence of 2.4%) [17]. Here, we have outlined a chronological study on the GLRaV-2 positive samples detected in a molecular diagnostic laboratory in South Australia since 2001. These detections occurred across different varieties and grape-growing regions of Australia. A special emphasis has been given to the phylogenetics of the virus in Australia, especially that of a rare variant of GLRaV-2 in *V. vinifera* cv. Grenache expressing a severe leafroll disease. 

## 2. Materials and Methods

### 2.1. Grapevine Material

A total of 11,257 grapevine samples of known *(n* = 1037) and unknown varieties (*n* = 10,220) were sent to our Lab for virus testing between 2001 and 2021 (Table 1 and Table 2). These samples were sent by growers to Waite Diagnostics, School of Agriculture, Food and Wine, The University of Adelaide, South Australia, for virus testing between 2001 and 2021 (Table 1). Unless otherwise stated, these grapevine samples were sent as dormant canes (Table 1) or as shoots in autumn when virus symptoms were visible (Table 2) [16]. A single sample came from either a single vine or consisted of a pool of five sub-samples (canes or shoots), each from a vine from the same block. Samples were sent in plastic bags at ambient temperature, and on arrival, they were stored at 4 °C prior to processing.

In 2004, five vines of cv. Chardonnay, grafted on Paulsen1103 with a decline, were selected in Victoria. At the same time, five grafted vines from the same vineyard were selected as negative control (Table 2). In addition, samples were collected from the Waite Research Institute (WRI) vineyard, including from a row of 50 vines of *V. vinifera*, cv. Grenache, clone SA137, growing on own roots and planted in 2005. As we demonstrated in the results (Section 3.1), the plants in this row were infected with GLRaV-2. Vines 2 and 5 (from the south) were samples for metagenomic NGS. The plants were inspected for symptoms after harvest when severe symptoms were at their peak. In 2021, one infected Chardonnay sample (clone OF) was obtained from Barossa Valley, South Australia (Chardonnay SA), and another unknown clone from Victoria (Chardonnay Vic). A sample of Red Globe table grape from the WRI vineyard was also collected for diagnosis. Samples were sent as shoots, and the status of their symptoms is given in Table 2. 

### 2.2. Total Nucleic Acid Extraction for RT-PCR

Total nucleic acid (TNA) from phloem scrapings of mature canes, trunks or green shoots and petioles (0.1–0.2 g) was extracted using Qiagen RNeasy Mini Kit (Hilden, Germany) following the manufacturer’s protocol [18]. Since 2006 (Table 1), TNA was extracted after absorption to a suspension of SiO2 (silica: Sigma-Aldrich, cat.S5631, Darmstadt, Germany) and elution with 120 µL 10 mM Tris, pH 8.5 following appropriate washing steps as described [19].

### 2.3. RT-PCR and Sanger Sequencing

All samples were tested for GLRaV-1, GLRaV-2, GLRaV-2-RG, GLRaV-3, GLRaV-4, GLRaV-4 strain 6 (4/6), GLRaV-4 strain 9 (4/9), GRSPaV, grapevine virus A (GVA), grapevine virus B (GVB), grapevine fleck virus (GFkV), and grapevine Pinot gris virus (GPGV; after 2016). Each virus was tested separately using one-step reverse transcription polymerase chain reaction (RT-PCR). The primers used in RT-PCR for the detection of viruses and virus variants are listed in Appendix A. Two sets of primers were used for the detection of GLRaV-2, LR2-U2/LR2-L2 and V2dCPf2/V2CPr1, while for the detection of the Red Globe (RG) strain of GLRaV-2, a specific pair (RGHSP227V/RGHSP777C) was used (Appendix A and Table 2). 

Each RT-PCR reaction contained Green Go Taq reaction buffer (Cat# M7911) (Promega, Madison, WI, USA), 0.2 µM each of the forward and reverse primer, 2.5 mM MgCl_2_, 0.2 mM of each dNTP (cat.72004) (Sigma-Aldrich, Darmstadt, Germany) and 10 mM DTT (Sigma-Aldrich, Cat# 43816, Darmstadt, Germany). In addition, EpiMark Hot Start Taq polymerase (NEB: New England Biolabs, Ipswich, MA, USA) and Protoscript II reverse-transcriptase (RT, NEB: New England Biolabs, Ipswich, MA, USA), Earlier PCR work was carried out in the presence of Superscript II RT and Platinum Taq polymerase (Invitrogen, Waltham, MA, USA). For each reaction, 9 µL of the RT-PCR reaction mixture and 1 µL TNA were added. 

The following RT-PCR thermocycling program was applied: 45 min at 46 °C, 2 min at 96 °C, followed by 35 cycles of 20 s at 94 °C, 20 s at 54–60 °C (depending on the primers), 30 s at 68 °C (72 °C for Platinum Taq Polymerase) and a final extension time of 5 min at 68 °C or 72 °C. RT-PCR products were resolved by electrophoresis in 1.5% agarose gel in TBE buffer (90 mM Tris-borate, 2 mM EDTA, pH 8.5) and visualised by UV light after staining with ethidium bromide.

Sanger sequencing was performed on RT-PCR amplicons (Australian Genome Research Facility, Adelaide, Australia) generated using the coat protein primers V2dCPf2 and V2dCPr1 [20] (Appendix A) for four isolates of GLRaV-2 as underlined in Table 2. Only two vines of Batch 34 (vine 2 and vine 5) were subjected to metagenomic HTS. To obtain a consensus sequence, at least three PCR fragments of each isolate obtained from the same PCR pair were sequenced in both directions.

### 2.4. Double-Stranded RNA Extraction

Samples for dsRNA extraction were collected in March 2021 from at least five randomly selected mature leaves across the whole canopy of two *Vitis vinifera* cv. Grenache plants (Vine 2 and Vine 5) (Appendix A). Three grams of petioles from the five leaves were extracted using the CF-11 method previously described [21]. The dsRNA was then precipitated with isopropanol and DNase treated prior to high-throughput sequencing (HTS).

### 2.5. Library Preparation and RNA Sequencing

A total of two libraries were prepared using facilities at our organisation in Melbourne, Australia. Each library was prepared using the TruSeq Stranded Total RNA with Ribo-Zero Plant kit (Illumina, San Diego, CA, USA), which performs ligation of the adapters and second-strand DNA synthesis. RNA concentration and integrity were determined using a NanoDrop ND-1000 (NanoDrop, Agilent, CA, USA), Tapestation (Agilent, CA, USA) and Qubit fluorometer (Thermo Fisher Scientific, Waltham, MA, USA) and confirmed by qPCR using KAPA Library Quantification Kit (Roche, Basil, Switzerland). Approximately 5 μg of total RNA was used to remove ribosomal RNA in accordance with the Ribo-Zero protocol. Following this, 2 × 150 bp paired-end sequencing on an Illumina Novaseq™ 6000 (Melbourne, Australia) was carried out following the manufacturer’s procedure.

### 2.6. Genome Assembly of Metagenomic Analysis

Illumina adapters were trimmed, and the reads with a quality score below 20 and length below 50 bp were removed using TrimGalore (v. 0.4.2) [22]. Trimmed reads were de novo assembled using SPAdes (v. 3.12.0) with default settings [23]. Assembled contigs were blasted against a local database containing all existing virus and viroid sequences from NCBI using BLAST+ (v. 2.11.0.) to obtain the virus status of each sample [24]. Later, Sanger sequencing using the CP primer pair V2dCPf2 and V2dCPr1 for GLRaV-2 [20] was used to confirm HTS sequence following single-tube RT-PCR. 

### 2.7. Phylogenetic and Genetic Diversity Analysis of the Sequences

Sequences from this study were submitted to GenBank, and accession numbers obtained (Appendix A). Publicly available sequences were downloaded from NCBI databases and aligned using Clustal W [25]. Thus, partial CP sequences of multiple GLRaV-2 isolates were considered following single tube RT-PCR starting from the initiation codon using the primer pair V2dCPf2/V2CPr1, and Sanger sequenced as described above. A phylogenetic tree for p24 was also generated using the public sequences of p24 in GenBank and the sequences p24 obtained by metagenomic HTS using vines 2 and 5 of cv. Grenache (Appendix A). Phylogenetic analysis was performed using the Neighbour-Joining method with 1000 bootstrap replicates by the MEGA (v. 7.0.26) software [26].

### 2.8. Recombination Analysis

GLRaV-2 contigs from Grenache Vines 2 and 5 with full-length or nearly full-length sequences were aligned with the sequences of the virus from GenBank (Appendix A) using Muscle (v. 3.8.31). The aligned sequences were trimmed according to the shortest sequence and analysed using RDP5 (v. Beta 5.23) [27]. The following seven methods: [28], GENECONV [29], Chimaera [30], MaxChi [31], BootScan [32], SiScan [33], and 3Seq [34] were selected for the analysis of recombination events. If at least four out of seven methods detected recombination event in a contig, the sequence was considered to be recombined, and it was excluded from further analysis.

## 3. Results

### 3.1. Incidence of GLRaV-2 in Australia

The results of virus testing on grapevine samples sent to our Lab since 2001 are summarised in Table 1 and Table 2. A total of 10,220 samples sent from unknown varieties were tested between 2001 and 2021 (excluding 2008). The name of the varieties was not given to us for confidentiality reasons. Of these samples, 170 (1.66%) tested positive for GLRaV-2 (Table 1). Most samples were sent for routine indexing (RI), but some growers were interested to know the disease status (DS) prior to top-working (TW). The samples were tested for 12 viruses, but only the test results for GLRaV-2 are shown in Table 1. The results for testing other viruses are shown in Table 2.

### 3.2. Detection of GLRaV-2 in Known Varieties

A total of 1037 samples of known varieties were sent by growers from 2001 to 2021. All the samples from the same variety in a given year were given the same batch number. A summary of RT-PCR testing for GLRaV-2 and eleven other viruses (see Section 2.3) is shown in Table 2. A total of 143 samples sent by growers (excluding our research samples which are underlined) tested positive for GLRaV-2 (13.7%). Overall, we detected GLRaV-2 in 18 known varieties. White varieties, including Chardonnay, were asymptomatic, except when GLRaV-2 infected Chardonnay was grafted on Paulsen showed decline (Figure 1). Of batches 5, 6, 8 and 18 (Table 2), only Chardonnay in Batch 6 tested negative for GLRaV-2 and was used as our negative control. Leafroll symptoms on red varieties, including cv. Emperor (Table 2, Batch 33), Grenache (Batch 34, Figure 2) and Shiraz (Batch 43, Figure 3) were also observed. All the samples tested positive for GLRaV-2 were also infected with at least one or more viruses. A sample of Chardonnay with a maximum virus load of 8 (Table 2, Batch 16) was still asymptomatic. Additionally, in one asymptomatic rootstock sample, Ramsey (Table 2, Batch 46), six different virus species were detected.

### 3.3. Symptoms

#### 3.3.1. Leafroll Symptoms

We observed leafroll symptoms associated with GLRaV-2 in two red varieties, Shiraz and Grenache (Table 2 and Figure 1 and Figure 2). Figure 1 shows a mild leafroll symptom of GLRaV-2 on Shiraz grafted onto cv. Viognier (Table 2, Batch 44). The asymptomatic grapevine rupestris stem pitting-associated virus (GRSPaV) was also present in this plant, but no other leafroll-associated virus was detected. 

*V*. *vinifera* cv Grenache, clone SA137, infected with GLRaV-2, displayed severe leafroll symptoms, which appeared after veraison and led to necrosis of the bottom leaves (Figure 2 and Table 2, Batch 34). In most leaves, the necrosis initiated from one side of the leaf and then progressed through the entire leaf (Figure 2A). These symptoms have been observed each year since 2016. A neighbouring row of cv. *Tempranillo*, which was positive for GRSPaV only and negative for GLRaV-2, did not show any symptoms. Both these varieties were planted in 2005.

#### 3.3.2. Graft Incompatibility Symptoms (GI)

In this study’s dataset, examples of GI (decline) were exclusively observed in Chardonnay grafted on Paulsen in 2003 (VIC), 2004 (VIC), 2005 (VIC) and 2019 (SA) (Table 2, Batch 2, 5, 8, and 20, respectively). As an example, decline was observed in 2004 in an established three-year-old commercial vineyard in VIC planted with cv. Chardonnay vines, grafted on Paulsen 1103. In GLRaV-2-infected Chardonnay, symptoms of GI appeared a year after grafting (Figure 3). Symptomatic samples tested positive for GLRaV-2 in all five vines (Batch 5, Table 2), whereas GLRaV-2 was not detected in samples from five asymptomatic vines (Batch 6, Table 2).

### 3.4. Genomic Studies of GLRaV-2

#### 3.4.1. Phylogenetic Analysis

A phylogenetic tree based on the partial nucleotide sequence of the CP gene (Figure 4A) and the full-length sequence of p24 (Figure 4B) was constructed using two Grenache vines, 2 and 5. The sequences were derived from either RT-PCR or HTS, as described in Appendix A. The following sequences of GLRaV-2 in Australia were used in the tree (Appendix A). Chardonnay SA (Aust) (OK334632), Chardonnay VIC (Aust) (OK334633) and Red Globe table grape from The University of Adelaide research vineyard were sequenced by Sanger sequencing. One full genome sequence of GLRaV-2 from Grenache vine 2 (OM179872) and two near full sequences from vine 5 (OK324337 and OM362846) in the same Grenache row were obtained by HTS and used in the tree to analyse both CP and p24 (Figure 4 A,B). OK324337 and OM179872 both belonged to the 93/955 phylogroup, while OM362846 belonged to the H4 phylogroup (Figure 4 and Appendix A). Although several contigs closely related to other GLRaV-2 PN group sequences were obtained from Grenache, no near full-length sequence could be assembled from the HTS data (not shown). Nevertheless, partial CP and full p24 sequences belonging to the PN clade were available and used in the phylogenetic analysis of Figure 4 A,B, respectively, confirming all three phylogroups were present in Grenache clone SA137.

#### 3.4.2. Full Virus Genomic Analysis

The phylogenetic tree is based on the full genomic sequence of three Australian GLRaV-2 isolates from cv. Grenache, one from vine 2 (OM179872) and two from vine 5 (OK324337, OM362846), shown in Figure 4C. These share a nucleotide identity between 85.47 and 99.9% with full viral genomic sequences in NCBI (Figure 4C and Appendix A). The highest pairwise identity (99.90%) amongst the three Australian cv. Grenache isolates was between GrenV2N20 and GREV5, both from the same vine row, which belonged to the 93/955 phylogroup. They also shared a high nt sequence identity (99.26–99.61%) with isolates from the USA, Canada, and South Africa. The third Australian isolate, GenV5N6H4 (OM362846), which coinfected the same vine, GREV5, shared the highest nucleotide identity (92.63 %) with an isolate from Brazil infecting *V. labrusca* cv. Isabel (KX774192).

#### 3.4.3. Recombination Detection

No recombination events were detected amongst the three Australian GLRaV-2 variants in cv. Grenache (OK324337, OM179872 and OM362846).

## 4. Discussion

A chronological study on the occurrence of GLRaV-2 in samples sent to a single molecular diagnostics laboratory in Australia over the past twenty years was carried out (Table 1 and Table 2). In a previous study, the incidence of the virus between 1998 and 2000 was reported as 2.4% [20]. As previously published, 2479 grapevine samples were tested between 1998 and 2000 [17], with GLRaV-2 detected in 61 of these samples (2.4%). The overall results from this study based on testing 11,257 samples from both known and unknown varieties were 313 positives (2.7%). This shows that the incidence of the virus was low (Table 1 and Table 2), confirming the previous results. This may indicate that the sanitary selection administered by Australian vine improvement associations following programmed testing of vines has been effective [35]. No apparent GLRaV-2 spread has occurred at the University of Adelaide research vineyard over 17 years, suggesting the absence of an active natural vector as outlined by others [9]. Since GLRaV-2 is not spreading naturally [12], it appears that the virus was originally present in cuttings imported to Australia, and spread occurred via propagation of this material or upon grafting. Viognier clone Montpellier is one accession that appears to be imported as infected cuttings [36].

Mixed virus infections were commonly observed in this study, especially in white varieties. Compared to red varieties (e.g., Shiraz), white varieties (e.g., Chardonnay and Viognier) can tolerate infection with multiple virus species (Table 2), including infection with GLRaV-2, without showing symptoms [36]. Chardonnay had the highest number of mixed infections among the white varieties (Table 2). For example, eight viruses (including GLRaV-2) were detected in a Chardonnay vine in 2012 that displayed no symptoms (Table 2, Batch 16). Viognier clone Montpellier 1968 is infected with GLRaV-2 and four other viruses [36], but it does not show symptoms. However, if it is top-worked with a sensitive red-berry variety, it will develop symptoms. Since this study commenced in 2001, the diversity of primers used for the detection of grapevine viruses has increased substantially. Therefore, we acknowledge that we missed detecting a few virus variants. This is especially true for GLRaV-3, which has an enormous range of virus variants which makes the detection of the virus by RT-PCR challenging. Mixed infections may be the result of a combination of viticultural practices, including top working to infected vines, blind selection of symptomless cuttings and acquisition of viruses through natural spread. Asymptomatic white varieties can act as inoculum sources and spread viruses via unhygienic viticultural practices.

Apart from GLRaV-2, two latent viruses, GRSPaV and grapevine rupestris vein feathering virus (GRVFV) (*Marafivirus*, *Tymoviridae*) [37], were detected in cv. Grenache. The severe leafroll symptoms in cv. Grenache was probably associated with GLRaV-2, as no other leafroll-associated virus was identified using either conventional RT-PCR or HTS (Table 2, Batch 34). In severely infected plants, older leaves turned necrotic. Grapevine leaf necrosis with this extent and severity has not been observed in leafroll-associated viruses, even with GLRaV-3, the most severe leafroll virus type [35,38]. The necrosis may result from a hypersensitive response [39] to infections associated with p24 (ORF 8) in GLRaV-2. The p24 gene (Figure 4B) is known to be an RNA silencing suppressor, and the eight amino acids required for its function, as reported, were detected here, confirming that p24 in Grenache is a functional suppressor of gene silencing [40,41]. In the citrus tristeza virus, another Closterovirus, the proteins encoded by three genes, p25, p20 and p23, act as silencing suppressors [42]. More studies are needed to see if other genes in GLRaV-2 can share the silencing suppressor function.

In 2004, GI was observed within 12 months of grafting Chardonnay on Paulsen 1103 (*V. berlandieri* x *V. rupestris*) (Table 2 and Figure 3). Since infection with GLRaV-2 culminates in a decline of grafted vines, the virus behaves similarly to a component of rugose wood complex [43]. The GI response of *V. vinifera* varieties on sensitive American hybrid rootstocks has been considered a hypersensitive necrotic response, a characteristic of certain GLRaV-2 isolates [44]. Several American hybrid rootstocks, with at least one parent of either *V. riparia* or *V. berlandei*, e.g., Kober 5BB, 3309 Couderc, 5C Teleki and 1103 Paulsen, show a hypersensitive reaction after being grafted with a GLRaV-2 infected budwood, which may lead to GI and a high mortality rate [41,45]. However, grafting of GLRaV-2 infected budwood onto tolerant rootstocks, such as 101-14 Mtg, or growing the infected material on its own roots, does not result in decline [44,45]. No hypersensitive reaction occurs when the rootstock is already infected with GLRaV-2 [45]. The reason for this is not known, but it may be related to the cross-protection phenomenon in plant viruses [46].

Of the six GLRaV-2 phylogroups, we found four in Australia: PN, 93/955, H4 and RG. The genomes of PN and 93/955 are more closely related to each other than to the genome of the RG group [12]. PN variants are widespread and induce both leafroll symptoms and graft incompatibility [12,47]. GLRaV-2 isolates detected in the Chardonnay samples from northern Victoria (OK334633, Appendix A) and South Australia (OK334632, Appendix A) were assigned to the PN group based on the partial nt sequences of CP (Figure 4A). The RG isolate from the USA (AF314061) has 99.8% homology with our RG isolate from cv. Red Globe based on its partial CP sequence obtained by Sanger sequencing.

The phylogenetic analysis using the HTS data obtained from the severe leafroll isolate of GLRaV-2 in the grapevine cv. Grenache (Figure 4) revealed the presence of three phylogroups, PN, 93/955 and H4, which was confirmed by Sanger sequencing of its CP gene (Appendix A and Figure 4A). This was further confirmed when the full-length nt sequence of p24 and the virus’s full genome was used in a phylogenetic analysis (Figure 4B,C). Phylogenetic analysis on the full-length sequence of GLRaV-2 of three isolates (Figure 4C) showed two of which were from phylogroup 93/955 and one from phylogroup H4. The identity of these matched with the respected group of each in the database. No full-length sequence of the PN group was found in our contigs, but partial sequences of this group were analysed in Figure 4A,B.

No recombination event was detected among these three phylogroups of GLRaV-2, which are present independently in the same vine (Figure 4). We investigated this by using the complete nt sequence of three isolates of GLRaV-2 in cv. Grenache in Australia, and searched for the recombination event. These isolates were GrenV2N20, which was found in Vine 2; isolate GrenV5N5 (OK324337); and isolate GrenV5N6H4 (OM362846), which co-infected the same grapevine (Vine 5, Appendix A).

In conclusion, although the incidence of GLRaV-2 is low in Australia, it can be a deleterious virus in two ways: firstly, it is associated with graft incompatibility, and secondly, it is associated with a severe leafroll disease, an example of which has been described here. Therefore, it must be considered an important debilitating virus, especially where grafting practice is desired or is mandatory, such as in phylloxera-infested regions where resistant rootstocks are essential.

## Figures and Tables

**Figure 1 viruses-15-01105-f001:**
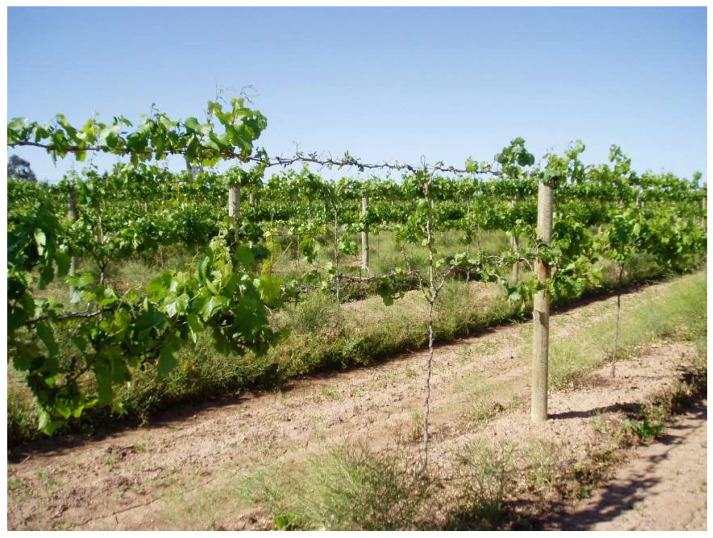
A vineyard of *Vitis vinifera* cv. Chardonnay grafted on Paulsen 1103 (Table 2, Batch 5). Vines infected with grapevine leafroll-associated virus 2 (GLRaV-2) show retarded growth and decline. The symptomatic vines were infected with GLRaV-2, and grapevine rupestris stem pitting-associated virus (GRSPaV), while healthy vines had only the inert virus, GRSPaV.

**Figure 2 viruses-15-01105-f002:**
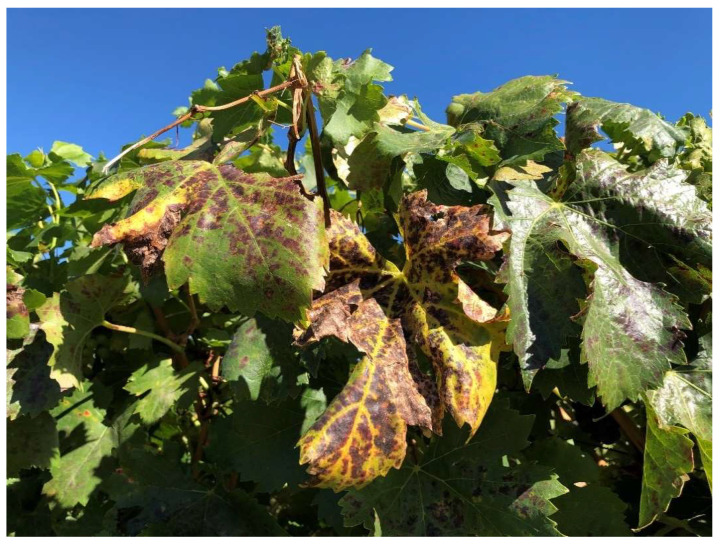
Severe symptoms of grapevine leafroll-associated virus 2 on *Vitis vinifera* cv. Grenache clone SA137. A severe leaf reddening and vein yellowing in late autumn. Note the necrosis of infected leaf starts from one side of the leaf, which will later expand to the whole leaf.

**Figure 3 viruses-15-01105-f003:**
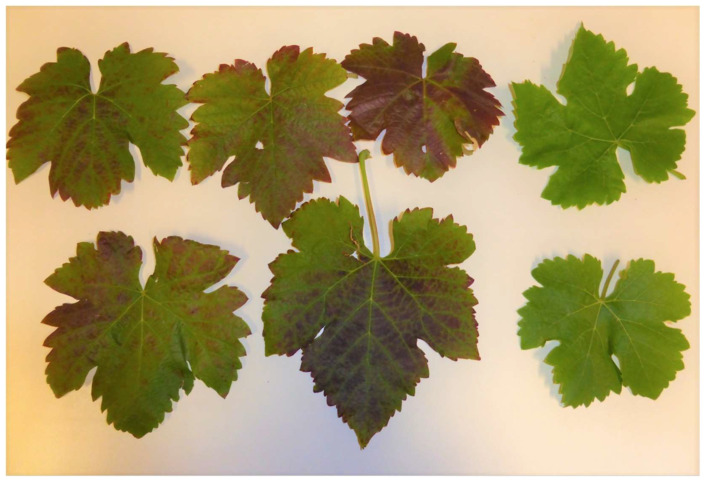
Mild leafroll symptoms on *Vitis vinifera* cv. Shiraz 1654 grafted on cv. Viognier tested positive for grapevine leafroll-associated virus 2 and grapevine rupestris stem pitting-associated virus (January 2015, Barossa Valley, South Australia). The pair of leaves on the right are healthy controls.

**Figure 4 viruses-15-01105-f004:**
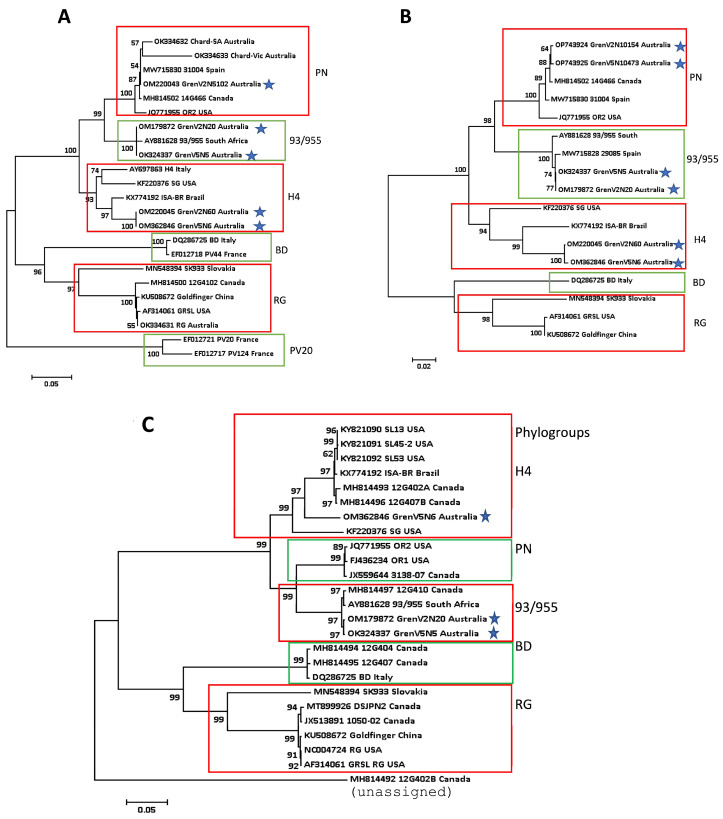
Phylogenetic mapping of the grapevine leafroll-associated virus 2 (GLRaV-2) variants from Australia and those selected from NCBI. Phylogenetic analysis was carried out using partial nt sequence of CP in Australian-grown Grenache, Chardonnay and Red Globe grapevine varieties. (**A**) The complete sequence of p24, (**B**) of selected isolates of GLRaV-2 in Grenache, and (**C**) nearly full-length virus genome from Vine 2 (V2) and Vine 5 (V5) of Grenache clone SA137. The sequences of the international isolates were obtained from the Genbank NCBI database. The p24 gene sequence of the PV20 phylogroup was not available in the NCBI database. The co-existence of three phylogenetic groups in V2 and V5 is shown by blue asterisks. Neighbour-Joining with bootstrap values of more than 50% (with 1000 repeats) was used for tree construction. Previously classified phylogenetic group names [12] are shown to the right of each clade. For further information on each contig, see Appendix A. For a clear comparative analysis, the outgroups were not used.

**Table 1 viruses-15-01105-t001:** Grapevine leafroll-associated virus 2 detected in samples sourced from unknown varieties and sent by Australian grapevine growers from 2001 to 2021.

Year	Total Samples Tested ^1^	Total Positives	% Positives	Reasons for Testing ^2^
2001	494	4	0.8	RL, TW
2002	109 ^2^	0	0	SS
2003	502	3	0.6	RI, DS
2004	1301	38	2.9	DS, RI, SS
2005	586	7	1.2	DS, RI, SS
2006	580	11	1.9	DS, RI, SS
2007	407	13	3.2	DS, RI, SS
2008	NA ^3^	NA	NA	NA
2009	473	5	1	SS, DS
2010	206	5	2.4	DS, TW
2011	254	2	0.8	DS, TW
2012	153 ^4^	13	8.5	DS, RI, SS
2013	260	4	1.5	DS, RI, SS
2014	133 ^4^	2	1.5	DS, RI, SS
2015	462	6	1.3	DS, RI, SS
2016	881	8	0.9	DS, RI, SS
2017	332	10	3	DS, RI, SS
2018	913	7	0.8	DS, RI, SS
2019	441	4	0.9	RI, TW, SS
2020	804	15	1.87	RI, TW, SS
2021	929	13	1.4	RI, TW, SS
Total	10,220	170	1.66	

^1^ Samples of various varieties sent by growers from South Australia (SA), Victoria (VIC), New South Wales (NSW), and Western Australia (WA). Only the test results for GLRaV-2 are shown here. ^2^ DS: Disease Status; RI: Routine Indexing; SS: Sanitary Selection; TW: Top-Working or top-grafting in the field. ^3^ Data not available. ^4^ These samples were sent to the Lab as symptomless green shoots.

**Table 2 viruses-15-01105-t002:** Results of RT-PCR testing of known grapevine varieties with recorded symptoms either sent by growers or from our research vineyard (2001 to 2021).

Batch #	Variety	State	*n1/n2* ^1^	Sampling Year	Symptoms ^2^	Other Viruses in Sample	Reason for Testing ^3^
1	Chardonnay	SA ^4^	2/257	2001	AS	RSP ^5^	DS, RI
2	Chard./Paulsen ^6^	VIC	2/12	2003	Decline	RSP	DS, RI
3	Chardonnay	VIC, NSW, SA	46/164	2004	AS	RSP, LR4/9, GVA	DS, RI, TW
4	Chardonnay-RG ^7^	VIC	9/43	2004	AS	RSP	RI, TW
5	Chard./Paulsen	VIC	5/5	2004	Decline	RSP	Research
6	Chard./Paulsen (H) ^8^	VIC	0/5	2004	AS (-ve control)	RSP	Research
7	Chardonnay	SA	1/1	2005	AS	LR1, LR3, GVA, RSP, FkV	RI, TW
8	Chard./Paulsen	VIC	4/7	2005	Decline	RSP	DS
9	Chardonnay	SA	1/6	2005	AS	LR1, RSP, FkV	DS, RI
10	Chardonnay	SA	8/22	2005	AS	RSP, FkV	DS, RI, TW
11	Chardonnay	SA	1/3	2006	AS	LR1, LR4/9, GVA, FkV	DS, RI, TW
12	Chardonnay (clone 96)	NSW	1/3	2006	AS	RSP	DS, RI
13	Chardonnay	SA	7/19	2007	AS	RSP, FkV	RI, TW
14	Chardonnay (grafted)	SA, VIC	2/3	2007	sLR	RSP, GFKV, GVA	RI, TW
15	Chardonnay	SA	2/6	2010	AS	RSP	DS, RI
16	Chardonnay	SA	1/1	2012	AS	RSP, LR1, LR4, LR4/5, LR4/9, GVA, FkV	DS, RI
17	Chardonnay	SA, NSW	12/48	2012	AS	GVA, GFkV, RSP	RI, TW
18	Chard./Paulsen	SA	2/2	2019	Decline	LR3, RSP	RI, SS, DS
19	Chardonnay	VIC	2/2	2021	AS	LR1, RSP	Research
20	Chardonnay, OF	SA	1/1	2021	AS	LR1, RSP	Research
21	Marsanne	SA	2/8	2003	AS	RSP	RI, TW
22	Marsanne-RG	VIC	2/8	2006	AS	RSP, GVA, FkV	RI, TW
23	Sauv Blanc	WA	1/45	2002	AS	RSP, LR4/9	RI, TW
24	Sauv Blanc	SA	2/27	2004	AS	LR4/9, RSP, GVA	RI, TW
25	Semillon-RG	WA	1/2	2005	AS	RSP	RI, TW
26	Semillon	SA	1/1	2010	AS	RSP	RI, TW
27	Viognier	VIC	1/3	2003	AS	RSP	DS
28	Viognier	SA	1/4	2006	AS	LR1, LR3, RSP, FkV	RI, TW
29	Viognier	SA	1/3	2007	AS	None	SS
30	Viognier	SA	1/3	2012	AS	RSP, GVA, FkV	TW
31	Viognier	SA	2/72	2015	AS	RSP	SS
32	Crimson Seedless	WA	1/16	2013	AS	RSP, GVA, FkV	TW
33	Emperor-RG	WA	1/16	2015	sLR	LR3, GVA, FkV, LR4-9	DS, RI, TW
34	Grenache SA137 ^9^	SA	3/3	2021	sLR	RSP, GRVF	Research
35	Grenache	SA	1/7	2021	AS	RSP, FkV	DS, RI, SS
36	Red Globe-RG	SA	10/10	2001	AS	RSP	Research
37	Merlot	SA	4/34	2002	AS	LR4/9, RSP	SS, TW
38	Merlot-RG	WA	2/6	2002	AS	RSP	DS, RI, SS
39	Nebbiolo	VIC	1/30	2002	AS	LR4/9, RSP, GVA, FkV	RI
40	Pinot Noir	SA	1/2	2021	AS	LR3, GVB	DS, RI, TW
41	Shiraz	VIC	4/15	2002	AS	RSP	AS
42	Shiraz	SA	1/4	2004	SD	GVA	DS
43	Shiraz	SA	2/2	2005	sLR	LR3, LR-4/5, RSP	DS, RI, SS
44	Shiraz/Viognier	SA	1/1	2015	mLR	RSP	DS, RI, TW
45	Ramsey-RG	SA	2/16	2002	AS	RSP, GVA, FkV	RI, SS
46	Ramsey	SA	1/16	2006	AS	LR-3, LR4/9, RSP, GVA, FkV	RI, SS
47	Ramsey	SA	1/5	2006	AS	RSP, GVA, FkV	RI, SS
48	Schwartzman-RG	VIC	1/16	2002	AS	RSP	RI, SS
49	Schwartzman	VIC	1/16	2004	AS	LR1, RSP	RI, SS
50	*V. rupestris*	VIC	1/9	2000	AS	LR1, RSP, GVA, FkV	RI, SS
51	101-14 Mtg	VIC	1/27	2004	AS	LR1	RI, SS

^1^ *n1/n2*: No. of positive samples/total samples tested within each batch. The overall *n1/n2* ratio for all batches (except for the underlined research batches) was 143/1037 (13.7%). ^2^ AS, asymptomatic, sLR, severe leafroll, mLR, mild leafroll. SD, Shiraz Disease: A GVA-associated disease in which plants show purple leaves and stunted growth in spring [2]. ^3^ DS: Disease Status; RI: Routine Indexing; SS: Sanitary Selection; TW: Top-Working or top-grafting in the field. ^4^ SA, South Australia; NSW, New South Wales; VIC, Victoria; WA, Western Australia. ^5^ RSP: Grapevine rupestris stem pitting-associated virus; GRVF: Grapevine rupestris vein feathering virus; LR-1, -3 or -4/9: Grapevine leafroll-associated virus 1, 3, 4/9; GLRaV-2-RG: RG Group of grapevine leafroll-associated virus 2; FkV: Grapevine fleck virus; GVA: Grapevine virus A. ^6^ Grafted on the Paulsen rootstock. ^7^ The RG group of GLRaV-2, which was detected by specific primers (Appendix A). ^8^ Healthy controls: five vines in Batch 6 growing in the same row as Batch 5 were used (see Section 3.3.2). ^9^ Grenache from our research vineyard, with known disease status, was used in meta-HTS.

## Data Availability

Representative sequences were deposited in GenBank under the accession numbers: OK324336, OK324337, OK334629, OK334631, OK334632, OK334633, OK662946, OK662946, OP743924, OP743925, OM179872 and OM362846.

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
