# Peer review of "A Chronological Study on Grapevine Leafroll-Associated Virus 2 in Australia"

_viruses, 2023, doi:10.3390/v15051105_

Round 1

Reviewer 1 Report

The manuscript of Habili and colleague provides a record of the temporal occurrence of grapevine leafroll-associated virus 2 (GLRaV-2) in Australia since 1999. It was found in 18 grapevine varieties, including Vitis rootstocks, and can be associated with various symptomology i.e., leafroll symptoms after veraison, graft incompatibility and vine decline in certain American hybrid rootstocks. Four out of the six GLRaV-2 phylogenetic groups were detected in Australia without any recombination events being measured/observed.

The manuscript is very informative since some of the historical data was not published before. The overall GLRaV-2 incidence is low and constant in time and the symptomology reported is mostly mild or asymptomatic, although, most of the plant tested were of white cultivars that do not generally express much symptoms.

The authors have combined multiple research data about GLRaV-2, including the overall evolution of the incidence of the samples processed by the Waite Diagnostics lab over time, symptom associations with GLRaV-2 detected in Australia, an experiment around graft incompatibility, high-throughput sequencing of a very symptomatic vine, assessment of the presence of the p24 protein and whether it was functional, and a recombination section. The fact that these points do not involve the same plants makes it difficult to read (and likely difficult to write).

The job of regrouping data collected over decades is not an easy one, as the technologies and knowledge of the target has changed considerably since 2001. For example, we now know that the primers used for the detection of GLRaV-3 in this work (LR3P-H420/C629, published in 1995), are working well for members of the phylogroup I but are not detecting most of the GLRaV-3 groups that have been described since.

I like the presentation of the virus detection over the time (Table 1) but I am wondering about the representativity of the samples tested? Why were the samples sent to the lab (disease response/certification process/material for propagation)? This is not discussed and to me, this would provide more information than the symptoms observed (since the assessment of the symptom is not described). If the information is available, it could be added to the table 2.

Also, the overall incidence of the samples tested is 1.7%, but the overall incidence of the samples identified (table 2) is 16%. I presume that the table 1 includes all the batches tested vs table 2 only include batches with at least one GLRaV-2. Can you confirm?

The authors claim that, for confidential reasons, we do not have access to the cultivar or the origin, but they do not provide the number of plants that this represents. I do not think that the cultivar alone or the state of origin would breach any confidentiality but I guess this information is not in the hands of the authors. Furthermore, the manuscript groups the different GLRaV-2 genetic groups based on the symptoms they cause. This concept is risky as the symptoms can vary greatly between cultivars and even between cultivar-rootstock combinations, as mentioned by the authors. Additionally, we do not know who recorded the symptoms, how/when they were recorded, or when the samples were collected. Furthermore, the sample tissue can be cane, presumably collected in winter.

Overall, the manuscript can be difficult to follow and the story is not focused. Some aspects are not explained well. When the authors refer to a molecular analysis, they should clarify that this is for the samples Sanger sequenced. The HTS section is still very unclear; it can be determined based on Supplementary Table 2 that this was done on two plants from the same cultivar (presumably with the same symptoms), but this should be stated explicitly in the &M and result section.

The relevance of the phylogenetic analyses of the P24 in this manuscript is still unclear. The work does not show anything new, the phylogeny is similar to the CP, and figure 5 is difficult to read (the difference between the amino acids in bold and the other is not easy to distinguish, and those bold aa are shared by all accessions). The manuscript mentions recombination analyses in the results and discussion, but no method is described. Additionally, the presence of GYSVd-1 should have been expected, as viroids are ubiquitous in vitis, so this should be explained in the manuscript.

There are many minor details that will need to be addressed:

In the abstract there is a mention of a spatial record, where the geographical origin of the samples is not discussed

L27: among viroids, ….. were detected

L92: could you give a number, instead of 90%

L95: is this the remaining 10%?

92-110: This paragraph would gain more explanation and distinction between the different groups of material (for different purpose).

L111: Is it total nucleic acid or total RNA?

L117: you should have only one ref here (the one you use)

L219: PCR mix: is the MgCL2, the final concentration or what is added to the mix? Where are the dNTPs?

L231: How many samples were chosen and how were they chosen?

L233: 3 PCR fragments? Are those replicates?

L239: dsRNA : which plant was/were selected? If it is only one plant, you should name it here too, to help the reader (From the plant Grenache 2 and Grenache 3, three grams of petioles from the five leaves were extracted using the CF-11 method previously describe)

L243: Where were the samples sequenced? How many libraries were prepared?

L285: the first sentence is not part of this study, and should be in the discussion only

L286: it would be nice to have this number in the material and method instead. How many were excluded due to confidentiality? Why is the different (10220) to the total obtained from table 2 (1234)

L300: When were the symptoms observed, and by whom (should be in the M&M). If the samples were sent with symptom description by the collector, this should be indicated (and taken with caution).

L302: “no symptoms were observed” yes, but those were white berry cultivars

L313: Table 2: when a GLRaV-2 genotype is described, are all the sample positive to the same genotype (e.g. 6- 9 chardonnay infected with GLRaV-2-RG)?

L313: the table would gain to have the total included

L313: What does a batch represent? Why were they tested in the first place? The size of the batch varies from 1 to 257. Could you identify the one tested individually vs the pool tests?

L317: Shiraz disease? Not described anywhere.

L327: why was a viognier used as rootstock (maybe to add in the discussion instead)

L328: GRSPaV : the letter P is missing

Author Response

  1. The manuscript of Habili and colleague provides a record of the temporal occurrence of grapevine leafroll-associated virus 2 (GLRaV-2) in Australia since 1999. It was found in 18 grapevine varieties, including Vitis rootstocks, and can be associated with various symptomology i.e., leafroll symptoms after veraison, graft incompatibility and vine decline in certain American hybrid rootstocks. Four out of the six GLRaV-2 phylogenetic groups were detected in Australia without any recombination events being measured/observed.

Reply: Thanks for the points. As mentioned in the text, recombination event between the phylogenetic groups were measured, but was not observed.

  1. The manuscript is very informative since some of the historical data was not published before. The overall GLRaV-2 incidence is low and constant in time and the symptomology reported is mostly mild or asymptomatic, although, most of the plant tested were of white cultivars that do not generally express much symptoms.

Reply: Thanks and corrected on page 63 by mentioning own-rooted white varieties don’t show symptoms. As described in the text, when Chardonnay is grafted on virus sensitive rootstocks the symptom of decline appears.

  1. The authors have combined multiple research data about GLRaV-2, including the overall evolution of the incidence of the samples processed by the Waite Diagnostics lab over time, symptom associations with GLRaV-2 detected in Australia, an experiment around graft incompatibility, high-throughput sequencing of a very symptomatic vine, assessment of the presence of the p24 protein and whether it was functional, and a recombination section. The fact that these points do not involve the same plants makes it difficult to read (and likely difficult to write).

Reply: We used two plants, Vine 2, and Vine 5,  of the same Grenache clone (SA137) from the same row (Table 2 and Table S2). Functional assessment of p24 of the GLRaV-2 isolate in V. vinifera cv. Grenache SA137 was not experimentally carried out, but its function in cv. Grenache was predicted based on phylogenetic analysis and sequence comparisons (Figures 4B and Figure 5), where  eight marker amino acids vital for the suppressor activity of p24 (derived from citations: [27] and [28]) were identified and shown by open rectangular boxes in Figure 5. For metagenomic analysis we did not rely on a single plant rather two plants. The GLRaV-2 nt in these two plants showed 99.9% homology (Figure 6). These plants have been described under dsRNA extraction in Materials and Methods.  

  1. The job of regrouping data collected over decades is not an easy one, as the technologies and knowledge of the target has changed considerably since 2001. For example, we now know that the primers used for the detection of GLRaV-3 in this work (LR3P-H420/C629, published in 1995), are working well for members of the phylogroup I but are not detecting most of the GLRaV-3 groups that have been described since.

Reply: Since 2001, we have been using this pair of primers for GLRaV-3, because no other pairs were as efficient. The RT-PCR results matched perfectly with the ELISA results. Since the main theme of our research was GLRaV-2, the metagenomic HTS on cv. Grenache did not show the presence of GLRaV-3 in our Grenache or any other research samples underlined in Table 2.

  1. I like the presentation of the virus detection over the time (Table 1) but I am wondering about the representativity of the samples tested? Why were the samples sent to the lab (disease response/certification process/material for propagation)? This is not discussed and to me, this would provide more information than the symptoms observed (since the assessment of the symptom is not described). If the information is available, it could be added to the table 2.

Reply: Thanks that you liked the data in Table 1. To answer your questions, the growers who sent samples to the Lab were interested to know the virus status (certification) or they were going to perform top working (grafting unwanted varieties in the field). The information has been added to page 95. Most samples send were dormant canes which had no symptoms in winter. Dormant canes were recommended as they gave a good PCR positive signal. Symptom description has already been added to Table 2 at the far-right column.

Also, the overall incidence of the samples tested is 1.7%, but the overall incidence of the samples identified (table 2) is 16%. I presume that the table 1 includes all the batches tested vs table 2 only include batches with at least one GLRaV-2. Can you confirm?

Reply: Overall the incidence of GLRaV-2 has been shown in Table 1 (1.7%), no matter if the variety in known or not, while Table 2 shows all known varieties tested positive for the virus (except negative control Chardonnay of batch 6). The reason is why the incidence of GLRaV-2 is higher than Table 1, is that in Table 2 we included positive samples for research purpose and these were underlined. Therefore, Table 2 does not tell us the natural incidence of the virus. Yes, we confirm that the positive samples were infected with at least one extra virus as shown in Table 2.

  1. The authors claim that, for confidential reasons, we do not have access to the cultivar or the origin, but they do not provide the number of plants that this represents. I do not think that the cultivar alone or the state of origin would breach any confidentiality but I guess this information is not in the hands of the authors. Furthermore, the manuscript groups the different GLRaV-2 genetic groups based on the symptoms they cause. This concept is risky as the symptoms can vary greatly between cultivars and even between cultivar-rootstock combinations, as mentioned by the authors. Additionally, we do not know who recorded the symptoms, how/when they were recorded, or when the samples were collected. Furthermore, the sample tissue can be cane, presumably collected in winter.

Reply: Yes, the confidentiality on revealing the name of varieties was not in our hands. On pages 100-101 the number of plants sent for testing was described. Out of 10220 samples (Table 1), 1034 samples were known varieties, as we mentioned in the text, the variety of majority of samples sent by growers was not known.

We did not use symptoms to create the phylogenetic trees, rather, we followed the phylogenetic grouping of Angelini et al (2017) [11] which was based on the nucleotide sequence homology. In fact, any symptom matched with a certain phylogenetic group was coincidental, and it was not in our hands, as depicted in Figure 4. Therefore, we did not group the GLRaV-2 isolates ourselves, rather we followed others. For our experiments we had access to our research vineyard at the University site (Table 2,  batches 34and 36 as well as the underlined Chardonnay samples). We already referred that most samples were sent as dormant canes.

  1. Overall, the manuscript can be difficult to follow and the story is not focused. Some aspects are not explained well. When the authors refer to a molecular analysis, they should clarify that this is for the samples Sanger sequenced. The HTS section is still very unclear; it can be determined based on Supplementary Table 2 that this was done on two plants from the same cultivar (presumably with the same symptoms), but this should be stated explicitly in the &M and result section.

Reply: Sanger sequencing was clarified and added to line 240. The HTS section was also clarified by adding the name of the vines (Vine 2 and vine 5) on lines 240 (in MM), and line 403 (in Results).

  1. The relevance of the phylogenetic analyses of the P24 in this manuscript is still unclear. The work does not show anything new, the phylogeny is similar to the CP, and figure 5 is difficult to read (the difference between the amino acids in bold and the other is not easy to distinguish, and those bold aa are shared by all accessions). The manuscript mentions recombination analyses in the results and discussion, but no method is described. Additionally, the presence of GYSVd-1 should have been expected, as viroids are ubiquitous in vitis, so this should be explained in the manuscript.

Reply: The phylogenetic analyses of the P24 gene confirms that of CP which show three independent phylogroups exist in vines 2 and 5. In fact the phylogroups of using p24 sequences have not been reported for GLRaV-2 before.

Method for recombination event has been added to MM  (see 2.7). Figure 5 has been made bigger and a dot system has been adopted.

GYSVd-1 and its symptoms have been further described in line 369.

There are many minor details that will need to be addressed:

  1. In the abstract there is a mention of a spatial record, where the geographical origin of the samples is not discussed.

Reply: In Table 2, column 3 the spatial distribution of the samples has been described: SA, South Australia; NSW, New South Wales; VIC, Victoria; WA, Western Australia.

  1. L27: among viroids, ….. were detected.

Reply: Done

  1. L92: could you give a number, instead of 90%

Reply: yes (n=7040)

  1. L95: is this the remaining 10%?

Reply: Fixed

  1. 92-110: This paragraph would gain more explanation and distinction between the different groups of material (for different purpose).

Reply: More info was added.

  1. L111: Is it total nucleic acid or total RNA?

Reply: Total nucleic acids

  1. L117: you should have only one ref here (the one you use).

Reply: Fixed, thanks for picking this up

  1. L219: PCR mix: is the MgCL2, the final concentration or what is added to the mix? Where are the dNTPs?
  2. Reply: Fixed
  3. L231: How many samples were chosen and how were they chosen?

Reply: A total of 10220 samples were tested (Table 1). As mentioned on the first paragraph of M&M, the samples were sent by growers.

  1. L233: 3 PCR fragments? Are those replicates?

Reply: Yes, fixed on page 245 of the amended version.

  1. L239: dsRNA : which plant was/were selected? If it is only one plant, you should name it here too, to help the reader (From the plant Grenache 2 and Grenache 3, three grams of petioles from the five leaves were extracted using the CF-11 method previously describe).

Reply: Fixed see Line 257

  1. L243: Where were the samples sequenced? How many libraries were prepared?

Reply: The answer for these were added to the text.

  1. L285: the first sentence is not part of this study, and should be in the discussion only

Reply: Thanks, this was fixed.

  1. L286: it would be nice to have this number in the material and method instead. How many were excluded due to confidentiality? Why is the different (10220) to the total obtained from table 2 (1037).

Reply: That sentence was moved to the M&M. Only the name of variety was not revealed for confidentiality reasons. Table 2 lists the samples of which the name of the relevant variety was revealed. So, 10220-1037 = 9183 this is the number of samples of which no variety name was known to us.

  1. L300: When were the symptoms observed, and by whom (should be in the M&M). If the samples were sent with symptom description by the collector, this should be indicated (and taken with caution).

Reply: The symptoms were recorded with caution by NH on the date of arrival/collection. This was mentioned in M&M under “Grapevine material”

  1. L302: “no symptoms were observed” yes, but those were white berry cultivars

Reply: You are right, thanks.

  1. L313: Table 2: when a GLRaV-2 genotype is described, are all the sample positive to the same genotype (e.g. 6- 9 chardonnay infected with GLRaV-2-RG)?

Reply: Each batch of Chardonnay had different profile of viruses (Table 2). For example, Batch 4 was infected with the RG strain of GLRaV-2.

  1. L313: the table would gain to have the total included

Reply: Total was added to the footnote.

  1. L313: What does a batch represent? Why were they tested in the first place? The size of the batch varies from 1 to 257. Could you identify the one tested individually vs the pool tests?

Reply: The definition of “batch is given in M&M (first paragraph). Vines from the same row, no matter if tested as pooled or as individual gave the same results, unless there was a suspension of a recent virus spread.

  1. L317: Shiraz disease? Not described anywhere.

Reply: Fixed (line 364 new draft).

  1. L327: why was a viognier used as rootstock (maybe to add in the discussion instead).

Reply: Added to L515 in the discussion.

  1. L328: GRSPaV : the letter P is missing

Reply: Fixed

Reviewer 2 Report

Manuscript viruses-2272705 reports the work conducted in a diagnostic lab in South Australia since 2001 to 2021, specifically work related to GLRaV-2, which is an economically important virus for grapevine production around the world. The paper presents interesting data, and the experimental design is correct, consequently I recommend publication. Below some suggestions/comments for the authors.

Line 20. Replace 1999 for 2001. In this paper you should report the new data generated starting in 2001. Data generated in 1999-2000 was already presented in a previous paper. Stick to 2001 and make the adjustments along the paper.

Line 20. …in 18 grapevine varieties and Vitis rootstocks…

Line 21. …different regions of Australia.

Line 22. …Grenache, was…

Line 62. Replace asymptomatic for absent or missing.

Line 77. Replace disease for symptoms.

Line 78. Replace variant for isolate.

Line 104. V. vinifera in Italic. 

Line 110. …was also collected for diagnosis.

Lines 219-222. After the manufacturer name add the country of origin, not the catalog number.

Line 237. Samples for dsRNA extraction were collected…

Line 241. …prior to high-throughput sequencing (HTS).

Line 247. (Agilent, USA) … (Thermo Fisher Scientific, USA)

Line 253. Genome assembly of metagenomic analysis

Line 258. Later, Sanger…

Line 259. …V2dCPr1 for GLRaV-2…

Lines 260-261. Delete this last sentence.

Line 263. Phylogenetic and genetic diversity analyses of the sequences.

Line 265. Publicly available sequences were downloaded…

Lines 266-268. Thus, partial CP sequences of multiple GLRaV-2 isolates were considered.

Lines 277-281. You also generated a phylogenetic tree from the p24, correct? You need to mention it.

Lines 285-286. This is not part of this study, delete.

Line 294. Change 1999 for 2001.

Line 314. Delete the reference.

Line 321. …control for GLRaV-2.

Line 326. Delete infected with GLRaV-2.

Line 342. You are still talking about the same Grenache vines. You should connect both paragraphs in one.

Line 358. …the symptoms of grapevine yellow speckle viroid 1… Do you have data supporting this sentence? If not, change the sentence.

376. Genomic studies of GLRaV-2

Figure 4. Not outgroup was included.

Line 466. Marafivirus, Tymoviridae in Italic.

Lines 479-484. Species names in Italic.

Line 480. Add a reference of a recent review about rugose wood disease.

Author Response

Line 20. Replace 1999 for 2001. In this paper you should report the new data generated starting in 2001. Data generated in 1999-2000 was already presented in a previous paper. Stick to 2001 and make the adjustments along the paper.

Reply: Done

Line 20. …in 18 grapevine varieties and Vitis rootstocks…

Reply: Done

Line 21. …different regions of Australia.

Reply: Done

Line 22. …Grenache, was…

Reply: Done

Line 62. Replace asymptomatic for absent or missing.

Reply: Done

Line 77. Replace disease for symptoms.

Reply: Done

Line 78. Replace variant for isolate.

Reply: Done

Line 104. V. vinifera in Italic. 

Reply: Done

Line 110. …was also collected for diagnosis.

Reply: Done

Lines 219-222. After the manufacturer name add the country of origin, not the catalog number.

Reply: Done

Line 237. Samples for dsRNA extraction were collected…

Reply: Done

Line 241. …prior to high-throughput sequencing (HTS).

Reply: Done

Line 247. (Agilent, USA) … (Thermo Fisher Scientific, USA)

Reply: Done

Line 253. Genome assembly of metagenomic analysis

Reply: Done

Line 258. Later, Sanger…

Reply: Done

Line 259. …V2dCPr1 for GLRaV-2…

Reply: Done

Lines 260-261. Delete this last sentence.

Reply: Done

Line 263. Phylogenetic and genetic diversity analyses of the sequences

Reply: Done

Line 265. Publicly available sequences were downloaded…

Reply: Done

Lines 266-268. Thus, partial CP sequences of multiple GLRaV-2 isolates were considered.

Reply: Done

Lines 277-281. You also generated a phylogenetic tree from the p24, correct? You need to mention it. Reply: Done

Lines 285-286. This is not part of this study, delete.

Reply: Done

Line 294. Change 1999 for 2001.

Reply: Done. Started from 2001 and fixed Table 2

Line 314. Delete the reference.

Reply: Done

Line 321. …control for GLRaV-2.

Reply: Done

Line 326. Delete infected with GLRaV-2.

Reply: Done

Line 342. You are still talking about the same Grenache vines. You should connect both paragraphs in one.

Reply: Done

Line 358. …the symptoms of grapevine yellow speckle viroid 1… Do you have data supporting this sentence? If not, change the sentence.

Reply: Done:  A reference was added.

  1. Genomic studies of GLRaV-2

Reply: Done

Figure 4. Not outgroup was included.

Reply: Unfortunately, when the outgroups were added,  the pattern of both trees changed and the new tree was hard to interpretate, we, then, decided to remove them. Instead, we added the following sentence to the Figure 4 caption: “For a clear comparative analysis, the outgroups have been deleted”.  We are waiting for your response.

Line 466. Marafivirus, Tymoviridae in Italic.

Reply: Done

Lines 479-484. Species names in Italic.

Reply: Done

Line 480. Add a reference of a recent review about rugose wood disease.

Reply: Done

Round 2

Reviewer 1 Report

The manuscript of Habili and colleague on grapevine leafroll-associated virus 2 (GLRaV-2) in Australia was slightly improved but many questions remained unanswered.

From the original review the Q3 (The authors have combined multiple research data …. makes it difficult to read), the answer is not satisfactory.

I do understand the response and I do understand why the different research were made at the time, but they don’t make an easy story to read when pilled up together. I suggested to remove the p24 study (a sentence in the discussion is sufficient e.g. the p24 is known to be a RNA silencing suppressor, and the eight amino acids required for its function as reported [26,27] were detected, confirming that p24 in Grenache is a functional suppressor of gene silencing).

The mention of GYSVd-1 symptoms is not necessary in this paper either

From the Q4 (The job of regrouping data ….. the GLRaV-3 groups that have been described since), the answer is not suitable either.

We know for a fact that those primers aren’t GLRaV-3 universal, and an acknowledgment of the authors would be nice (and I believe it is the same for some of the other primers). I know that this is not the aim of this work but they should mention that the knowledge of the diversity for those viruses have increased tremendously since the beginning of the study and it is possible that some detections were missed.

Btw, the authors referred to the ELISA results in the answer. ELISA was not used in the ms, and we know that some serotypes of GLRaV-3 are/were not well detected by ELISA.

From the Q5 (I like the presentation of the virus detection over the time (Table 1) …. it could be added to the table 2) the authors did not respond well.

The representativity of the sample is not discussed once. The reason why samples were sent (certify/top-grafting/…) is the information I’d like to see in table 1 (when known) as this could explain the phytosanitary status of the material. The type of tissue tested should also be added in the table, since no symptoms can be seen in dormant wood (This suggest that all samples in table 2 were shoots/leaves)  

For the rest, authors responded to many comments as “fixed” but I could not find this in the ms (unless I have the wrong version):

“The manuscript mentions recombination analyses in the results and discussion, but no method is described”. (only the ref is added) this need to be clarified

“GYSVd-1 and its symptoms have been further described in line 369”.

no the symptoms are not described

“In Table 2, column 3 the spatial distribution of the samples has been described”

the origin of the sample is indicated, but it has not been described, analyzed nor discussed. I suggest to remove "spatial" from the abstract as it is misleading (once again the representativity of the samples is not discussed)

“L92: could you give a number, instead of 90% Reply: yes (n=7040)”

there is no 7040 in the ms

“L95: is this the remaining 10%? Reply: Fixed”

where?

“L117: you should have only one ref here (the one you use). Reply: Fixed, thanks for picking this up”

on the old version there were 3 ref, there are now 4 on the new one

“L243: Where were the samples sequenced? How many libraries were prepared? Reply: The answer for these were added to the text”

the number of libraries was completed, not the facility where the samples were sequenced 

“L317: Shiraz disease? Not described anywhere. Reply: Fixed (line 364 new draft)”

I cannot see any description of SD

In addition, I’d like to point out:

I missed it in the first review, but no information is given about the HTS data (output, read length, single/paired) and the number of contigs obtained, the length and coverage.

Should the HTS data made public?

Many reference are wrong, the author should check all the numbers (at least ref 19 L469 and 471, ref 37 L508, ref 42 L517)

L22: the clone of the Grenache should be listed in its first description

L52: the transmission of closterovirus is not known for all of them (I cannot find the vector for Rehmannia virus 1 or cnidium closterovirus 1)

L63: “own-rooted white cultivar”: symptoms of GLRaV-2 are generally not visible in most white cultivar even when grafted

L79-80: “but not when the vines are on their own roots” – there shouldn’t be any grafts when the vines are own-rooted so no GI is expected

L83: it is easier to state the incidence obtained previously here than to force reader to fetch it :(

L98: “Samples with known varieties were batched according to the year sent and the variety”. What does that mean? Were all the sample from the same variety processed together on a given year?

L103: when samples were shoots/leaves only

L111-115: why are those one better described than the rest? no sympt described here everything is in the table 2, if you list those, you need to describe all or maybe state why those were selected for the sanger?

L123: only one ref

L242: “for a number of GLRaV-2 isolates” how many?

L250: “of two Vitis vinifera cv. Grenache plants (Vine 2 and Vine 5)” instead

L264: on an Illumina Novaseq™ 6000 – where?

L270: give a reference for the tools used (Spades and TrimGalore)

L281: “Table S2” this is only the accessions from the HTS Grenache

L 305: I guess this is where you should give the program used instead of a ref (Recombination Detection Program RDP-5 (v5.5) with the ref of the tool and the parameters settings

L337-340: I don’t understand the need of this paragraph (especially the last sentence)

Table 2: the negative control Chardonnay/Paulsen shouldn’t be underlined (I guess the virus was not sequenced from it)

L347: SD still undescribed anywhere

L372: what are the symptoms of GYSVd-1 and why are they only visible on 10 %. Is this necessary in this paper?

L395: “The infected vines had GLRaV-2…” the symptomatic plants were infected with

L427: “For a clear comparative analysis, the outgroups have been deleted” deleted or not used in the analyses?

Figure 5 : clearer but not needed as the aligned sequences all have the required aa (could be replaced by a sentence in the discussion)

Figure 6: a phylogenetic analysis on full genome is more useful (easier to read and interpret)

L469: it would have been nice to discuss the representativity of the sample tested (% of different varieties and region compared with the grape-growing regions of Australia, difference of GLRaV-2 incidence in the different area and varieties, ….)

L490: “super spreader” is not the right concept here, especially for a vector-less virus

L520: no GI can be expected from self-rooted grapes

L535: figure 6 does not illustrate the absence of recombination events

L535-537: the sentence is not clear. Why is there only one accession number and which isolate is in which plant?

L538-540: distance matrix …. Not important

Author Response

Dear editor

The following is our point-by-point answer to the Reviewer 1 points raised in the second round of the review:

Our answers are highlighted here and the paragraph number on the manuscript was given.

The manuscript of Habili and colleague on grapevine leafroll-associated virus 2 (GLRaV-2) in Australia was slightly improved but many questions remained unanswered.

From the original review the Q3 (The authors have combined multiple research data …. makes it difficult to read), the answer is not satisfactory.

Author: We tried to make it clearer. Reviewer 1 wrote “we combined multiple research data?” Please refer to the multiple research data then we will try to segregate them.

I do understand the response and I do understand why the different research were made at the time, but they don’t make an easy story to read when pilled up together. I suggested to remove the p24 study (a sentence in the discussion is sufficient e.g. the p24 is known to be a RNA silencing suppressor, and the eight amino acids required for its function as reported [26,27] were detected, confirming that p24 in Grenache is a functional suppressor of gene silencing).

As suggested by the Reviewer 1, the study of p24 was removed and just a sentence on lines 989-993 was added (see also line 871 confirming the removal).

The mention of GYSVd-1 symptoms is not necessary in this paper either

Author: Notes related to GYSVd-1 have been removed.

From the Q4 (The job of regrouping data ….. the GLRaV-3 groups that have been described since), the answer is not suitable either.

We know for a fact that those primers aren’t GLRaV-3 universal, and an acknowledgment of the authors would be nice (and I believe it is the same for some of the other primers). I know that this is not the aim of this work but they should mention that the knowledge of the diversity for those viruses have increased tremendously since the beginning of the study and it is possible that some detections were missed.

Author: GLRaV-3 and 10 other viruses were not the main theme of this work. We described the diversity of recently published primers by adding a new sentence on lines1175-1179 in “Discussion” which is generic for all grapevine viruses we tested in our study.

Btw, the authors referred to the ELISA results in the answer. ELISA was not used in the ms, and we know that some serotypes of GLRaV-3 are/were not well detected by ELISA.

Author: This sentence about ELISA was used only in response to the referee’s report, as we never discussed ELISA for any virus in our MS. We just used RT-PCR for virus detection.

From the Q5 (I like the presentation of the virus detection over the time (Table 1) …. it could be added to the table 2) the authors did not respond well.

The representativity of the sample is not discussed once. The reason why samples were sent (certify/top-grafting/…) is the information I’d like to see in table 1 (when known) as this could explain the phytosanitary status of the material. The type of tissue tested should also be added in the table, since no symptoms can be seen in dormant wood (This suggest that all samples in table 2 were shoots/leaves)  

Author: Tables 1 and 2 must be kept separate: Table 1 lists samples with unknown variety names, which were sent over 20 years (total n = 10220). Apart from naming the state, no other information was given. Table 2 lists 1037 samples with variety names. In order to describe the grower’s reason for sending samples, we added a column to each Table (Table 1 line 490, Table 2 line 780).to explain whether the samples were sent for routine indexing, sanitary selection, top-working or disease status. Other specifications of samples were not given to us.

For the rest, authors responded to many comments as “fixed” but I could not find this in the ms (unless I have the wrong version):

Author: We fixed point by point, but we are not sure what had happened.

“The manuscript mentions recombination analyses in the results and discussion, but no method is described”. (only the ref is added) this need to be clarified

Author: The recombination method has been added to the Methods under section 2.8

“GYSVd-1 and its symptoms have been further described in line 369”.

no the symptoms are not described

Author: As recommended by Reviewer 1, mentioning of the GYSVd-1 symptoms has been deleted.

“In Table 2, column 3 the spatial distribution of the samples has been described”. The origin of the sample is indicated, but it has not been described, analyzed nor discussed. I suggest to remove "spatial" from the abstract as it is misleading (once again the representativity of the samples is not discussed)

Author: The term spatial is removed. The samples were sent by growers for commercial virus testing. No additional information was provided. Revealing the GPS or the winery name where the samples were sent from for diagnosis is against the rule of the organisation, where client confidentiality must be maintained. However a line has been added in the discussion to acknowledge that GLRaV-2 occurs in grape growing regions in various states across the country.

“L92: could you give a number, instead of 90% Reply: yes (n=7040)”

there is no 7040 in the ms

“L95: is this the remaining 10%? Reply: Fixed”

where?

Author:  90% and 10% have been deleted. Sorry, 7040 was a typo: the correct number for total samples is written under the footnote of each Table (1 & 2).

“L117: you should have only one ref here (the one you use). Reply: Fixed, thanks for picking this up” on the old version there were 3 ref, there are now 4 on the new one

Author: only one reference [17] has been added under: 2.2. Total Nucleic Acid Extraction for RT-PCR (third line).

“L243: Where were the samples sequenced? How many libraries were prepared? Reply: The answer for these were added to the text”

the number of libraries was completed, not the facility where the samples were sequenced 

Author: Sanger sequencing of PCR products was done at: Australian Genome Research Facility, Adelaide, Australia (section 2.3 last paragraph). High throughput sequencing was done in-house, at AgriBio, the facility of one of the authors (Constable), on a Novaseq and the facility does not require to be listed

The number of libraries were two and added to section: 2.5. Library preparation and RNA sequencing

“L317: Shiraz disease? Not described anywhere. Reply: Fixed (line 364 new draft)”

I cannot see any description of SD

Author: Shiraz Disease was described in line 2 of the footnote of Table 2. The reference [2] has been allocated for SD.

In addition, I’d like to point out:

I missed it in the first review, but no information is given about the HTS data (output, read length, single/paired) and the number of contigs obtained, the length and coverage.

Author: This information has been added to Table S2

Should the HTS data made public?

Author: No, in Australia newly developed standards suggest only the consensus sequence should be uploaded. If required, the raw sequences for GLRaV-2 can be uploaded from the samples. The remaining sequence reads, which are not analysed are not relevant to the manuscript.

Many reference are wrong, the author should check all the numbers (at least ref 19 L469 and 471, ref 37 L508, ref 42 L517)

Author: We have reorganised the references in order and double-checked.

L22: the clone of the Grenache should be listed in its first description

Author: The clone name “SA137” has been added.

L52: the transmission of closterovirus is not known for all of them (I cannot find the vector for Rehmannia virus 1 or cnidium closterovirus 1)

Author: This has been amended on the last line of paragraph two in “Introduction”

L63: “own-rooted white cultivar”: symptoms of GLRaV-2 are generally not visible in most white cultivar even when grafted

Author: If the scion, for example Chardonnay, is grafted on virus sensitive rootstocks, like Kober 5BB, it will show graft incompatibility. (See: [44]). We have changed it to “white varieties” in the second line of paragraph 4 of “Introduction”.

L79-80: “but not when the vines are on their own roots” – there shouldn’t be any grafts when the vines are own-rooted so no GI is expected

Author: Yes own rooted vines (ungrafted vines) don’t show GI symptoms, see [14]

L83: it is easier to state the incidence obtained previously here than to force reader to fetch it :(

Author: The incidence of the virus has been added to the last paragraph of “Introduction”

L98: “Samples with known varieties were batched according to the year sent and the variety”. What does that mean? Were all the sample from the same variety processed together on a given year?

Author: That’s correct and, in our opinion, the matter was resolved by adding a sentence in lines 1 and 2 of the first paragraph in section: 3.2. Detection of GLRaV-2 in known varieties.

L103: when samples were shoots/leaves only

Author: Table 2 deals with samples of known variety and known tissue type

L111-115: why are those one better described than the rest? no sympt described here everything is in the table 2, if you list those, you need to describe all or maybe state why those were selected for the sanger?

Author: We have an easy access to our research vineyard which contains a variety collection block. We know the virus profile of selected vines.

L123: only one ref

Author: One was deleted (highlighted)

L242: “for a number of GLRaV-2 isolates” how many?

Author: four isolates (see last paragraph of the 2.3 section)

L250: “of two Vitis vinifera cv. Grenache plants (Vine 2 and Vine 5)” instead

Author: This has been changed

See: 2.4 Double-Stranded RNA extraction lines 2-3

L264: on an Illumina Novaseq™ 6000 – where?

Author: This has been added to the paragraph with the heading:

2.5. Library preparation and RNA sequencing

L270: give a reference for the tools used (Spades and TrimGalore)

Author: the reference was given

L281: “Table S2” this is only the accessions from the HTS Grenache

Author: This has been corrected under

2.7. Phylogenetic and genetic diversity analysis of the sequences

L 305: I guess this is where you should give the program used instead of a ref (Recombination Detection Program RDP-5 (v5.5) with the ref of the tool and the parameters settings

Author: Details were given in 2.8. Recombination analysis

L337-340: I don’t understand the need of this paragraph (especially the last sentence)

Author: The paragraph under “3.2. Detection of GLRaV-2 in known varieties” was removed

Table 2: the negative control Chardonnay/Paulsen shouldn’t be underlined (I guess the virus was not sequenced from it)

Author: The underline was removed in Table 2

L347: SD still undescribed anywhere

Author: Added to footnote of Table 2

L372: what are the symptoms of GYSVd-1 and why are they only visible on 10 %. Is this necessary in this paper?

Author: This study on GYSVd-1 has been removed from this paper.

L395: “The infected vines had GLRaV-2…” the symptomatic plants were infected with

Author: Corrected in the caption of Figure 3.

L427: “For a clear comparative analysis, the outgroups have been deleted” deleted or not used in the analyses?

Author: Corrected and changed to “were not used”

Figure 5 : clearer but not needed as the aligned sequences all have the required aa (could be replaced by a sentence in the discussion)

Author: Figure 5 has been removed and replaced by a sentence in “Discussion”.

Figure 6: a phylogenetic analysis on full genome is more useful (easier to read and interpret)

Author: Figure 6 has been removed and replaced with a full-length tree as Figure 5C.

L469: it would have been nice to discuss the representativity of the sample tested (% of different varieties and region compared with the grape-growing regions of Australia, difference of GLRaV-2 incidence in the different area and varieties, ….)

Author: Since a total 10,220 samples were from unknown varieties it was not possible to gain a comprehensive representativity. Only 1,037 samples had their variety tagged, but these were not enough for analysis. A line has been added to the discussion to say that GLRaV-2 was found in various varieties and in all states from which samples were sent.

L490: “super spreader” is not the right concept here, especially for a vector-less virus

Author: this phrase has been removed. See the last sentence of paragraph 2

L520: no GI can be expected from self-rooted grapes

Of course, and the sentence has been modified as shown in paragraph 4 of the discussion.

L535: figure 6 does not illustrate the absence of recombination events

Author: Figure 6 has been removed

L535-537: the sentence is not clear. Why is there only one accession number and which isolate is in which plant?

Author: the sentence was modified to make it clearer using the accession numbers (view in Discussion, one paragraph before last).

L538-540: distance matrix …. Not important

Author: this was removed

Submission Date

24 February 2023

Date of this review

05 Apr 2023 14:50:06
